# Aryl Hydrocarbon Receptor in Post-Mortem Hippocampus and in Serum from Young, Elder, and Alzheimer’s Patients

**DOI:** 10.3390/ijms21061983

**Published:** 2020-03-14

**Authors:** Nicte Alaide Ramos-García, Marisol Orozco-Ibarra, Enrique Estudillo, Guillermo Elizondo, Erick Gómez Apo, Laura Graciela Chávez Macías, Ana Luisa Sosa-Ortiz, Mónica Adriana Torres-Ramos

**Affiliations:** 1Unidad Periférica de Neurociencias, Instituto Nacional de Neurología y Neurocirugía/Universidad Nacional Autónoma de México. Av. Insurgentes Sur No. 3877 Col. La Fama, Tlalpan, C.P. 14269, Ciudad de México, Mexico; alaidenic2479@gmail.com; 2Departamento de Biología Celular, CINVESTAV-IPN, Av. Instituto Politécnico Nacional No. 2508, Col. San Pedro Zacatenco, Gustavo A. Madero, C.P. 07360, Ciudad de México, Mexico; gazuela@cell.cinvestav.mx; 3Laboratorio de Neurobiología Molecular y Celular, Instituto Nacional de Neurología y Neurocirugía. Av. Insurgentes Sur No. 3877 Col. La Fama, Tlalpan, C.P. 14269, Ciudad de México, Mexico; marisol.orozco.ibarra@gmail.com; 4Laboratorio de Reprogramación Celular, Instituto Nacional de Neurología y Neurocirugía/Universidad Nacional Autónoma de México. Av. Insurgentes Sur No. 3877 Col. La Fama, Tlalpan, C.P. 14269, Ciudad de México, Mexico; jestudilloh@gmail.com; 5Hospital General de México, “Dr. Eduardo Liceaga”. Dr. Balmis No. 148, Col. Doctores, Cuauhtémoc, C.P. 06720, Ciudad de México, Mexico; erickapo@hotmail.com (E.G.A.); laurachm@prodigy.net.mx (L.G.C.M.); 6Laboratorio de Demencias, Instituto Nacional de Neurología y Neurocirugía, Av. Insurgentes Sur No. 3877 Col. La Fama, Tlalpan, C.P. 14269, Ciudad de México, Mexico; drasosa@hotmail.com

**Keywords:** AHR, ARNT, astrocytes, microangiopathy, neuroinflammation, gliosis

## Abstract

One of the characteristics of the cerebral aging process is the presence of chronic inflammation through glial cells, which is particularly significant in neurodegeneration. On the other hand, it has been demonstrated that the aryl hydrocarbon receptor (AHR) participates in the inflammatory response. Currently, evidence in animal models shows that the hallmarks of aging are associated with changes in the AHR levels. However, there is no information concerning the behavior and participation of AHR in the human aging brain or in Alzheimer’s disease (AD). We evaluated the expression of AHR in human hippocampal post-mortem tissue and its association with reactive astrocytes by immunohistochemistry. Besides this, we analyzed through ELISA the AHR levels in blood serum from young and elder participants, and from AD patients. The levels of AHR and glial fibrillar acid protein were higher in elder than in young post-mortem brain samples. AHR was localized mainly in the cytosol of astrocytes and displayed a pattern that resembles extracellular vesicles; this latter feature was more conspicuous in AD subjects. We found higher serum levels of AHR in AD patients than in the other participants. These results suggest that AHR participates in the aging process, and probably in the development of neurodegenerative diseases like AD.

## 1. Introduction

The aryl hydrocarbon receptor (AHR) is a ligand-activated transcription factor widely known for its role in mediating the detoxification of xenobiotics. In the absence of its ligands, AHR remains inactive and forms a complex with the chaperone heat shock protein 90, p23, and AHR-associated protein, whose primary function is to retain AHR in the cytoplasm. When AHR interacts with its ligands, it undergoes conformational changes that expose its nuclear localization sequence, thus promoting AHR translocation to the nucleus, where it heterodimerizes with the AHR nuclear translocator (ARNT). The complex formed by the AHR–ligand–ARNT binds to specific DNA regions called xenobiotics response elements or dioxin response elements located in the promoter regions of their target genes, to regulate their expression [1]. In addition to its role in detoxification, AHR is involved in the circadian cycle and glucose metabolism [2], retinoid homeostasis [3], differentiation and cell division [4], memory and neurogenesis [5], protein degradation via the ubiquitin–proteasome system [6], senescence [7], and inflammation [8]. Besides this, it has been shown that AHR participates in the regulation of the innate and adaptive immune response [1,9]. AHR-knockout mice exhibit characteristics linked with inflammation in glia and neurons, like the demyelination of optic nerve [8] and microglial accumulation in the retina [10]. In line with this evidence, Rothhammer et al. [11] described that AHR attenuates the production of pro-inflammatory interleukins by astrocytes and microglia after stimulation with type I interferons. Additionally, Rothhammer et al. [12] reported that AHR activation regulates the expression of transforming growth factor alpha (TGF-α) and vascular endothelial growth factor B (VEGF-B) respectively in human microglia, which modulate the pro-inflammatory response in astrocytes. The involvement of AHR in the modulation of inflammation also suggests its participation in the chronic low-grade inflammation (so-called “inflammaging”), which is associated with aging and neurodegenerative diseases. Regardless of the confirmed presence of the AHR in human brain cells [11,12], the information about its role in the human nervous system and its associated pathologies is still limited.

Aging is a time-related decline in biological functions, which ultimately results in organismal death [13]. It is mainly accompanied by a shift within innate immunity toward a pro-inflammatory status. Therefore, during aging, a chronic and generalized inflammation in the central nervous system occurs, especially in brain regions such as the hippocampus. In neurodegenerative diseases, neuroinflammation is higher than in healthy elder brains. Thus, aging is one of the foremost risk factors for neurodegeneration, such as Alzheimer’s disease (AD), the principal neurodegenerative disease and the most common form of dementia (50%–56% of total cases). Glial cells include microglia and astrocytes, which play a dual role by amplifying the effects of inflammation and the mediation of cell damage, and by the protection of the nervous system. They undergo significant changes in aging and AD, namely the permanent activation of microglia and the increase in glial fibrillary acidic protein (GFAP), which is known as astrogliosis and is a hallmark of reactive astrocytes. The above promotes a chronic inflammation that favors the loss of homeostasis and predisposes to neuronal death [14]. These effects impair the structure and neuronal function during the pathogenesis of neurodegenerative diseases, such as AD and other dementias.

At present, evidence in different cell lines [15] and animal models [16,17] shows that the hallmarks of cell senescence and aging are associated with changes in the AHR levels. Therefore, taking into account the central role that AHR has in inflammation, and the scarce information about human AHR in conditions characterized by chronic low-grade inflammation, like aging and Alzheimer’s disease, in the present study, we evaluated the changes in the expression of AHR and GFAP in human hippocampal post-mortem tissue from young and elder subjects. Likewise, we assessed AHR levels in serum from young and elder participants, as well as Alzheimer’s patients. We found that AHR participates in the aging process and probably in the development of AD through the response of astrocytes, which can release AHR into microvesicles.

## 2. Results

### 2.1. The Aryl Hydrocarbon Receptor Expression is Higher in the Elder Than in the Young Human Hippocampus 

We confirmed the hippocampus integrity through hematoxylin–eosin staining (Figure 1A). We found distinctive gliosis morphological changes in astrocytes and an increase of GFAP expression in elder participants, when compared with young ones (*t* = 2.537, df = 11, p = 0.0276, Figure 1B,C).

The antibody against AHR was previously tested in C57BL/6 mouse liver tissue to evaluate its effectiveness in immunohistochemistry (Figure 2). We found a significant increase of AHR expression in elder participants when compared with young participants, in the hippocampus (*t* = 2.616, df = 11, p = 0.024, Figure 3A,B); the AHR stain was predominantly in the cytoplasm of non-neuronal cells (t = 23.53, df = 10, Figure 3C). We did not find differences between the sexes in these samples (data not shown). The immunofluorescence staining of the AHR antibody has been previously observed [18]. We found that AHR in elder post-mortem tissue predominantly colocalized with the glial marker GFAP for astrocytes (Figure 4A), and was mainly located in astrocytes’ cytoplasm, with an apparent morphology of being contained in extracellular microvesicles. We compared the levels of AHR expression in post-mortem tissue of elder vs. AD patients, and found an apparent increase of AHR in AD patients, and interestingly, in the samples from AD patients, we found an intense staining with a vesicular pattern in all the areas examined, as well as outside of some astrocytes (Figure 4B). 

### 2.2. The circulating Levels of Aryl Hydrocarbon Receptor Are Higher in AD Than in Young and Elderly Participants 

As AHR was almost undetectable in blood serum and considering the possibility that AHR could be present in extracellular vesicles, we treated the serum samples with 4% Tween 20, a reported way to break circulating exosomes [19]. In these detergent-treated samples, AHR was detectable, and we found no differences between the AHR of young and elder participants (*t* = 1.006, df = 28, p = 0.3232, Figure 4A). However, we found a significant increase in AHR concentration in the serum of AD patients, when compared with elder participants (*t* = 2.621, df = 43, p = 0.0121, Figure 5A). Strikingly, we identified microangiopathy as a condition that decreases the AHR level in AD patients (*t* = 2.151, df = 27, p = 0.04, Figure 5B). Considering the effect that we observed with microangiopathy, we decided to carry out subsequent analyses only with AD patients without this comorbidity. In this way, the expression of AHR was dramatically increased in AD patients compared with the elderly (t = 3.671, df = 33, p = 0.0008, Figure 6A), and the AHR levels in AD patients were not associated with disease severity (F_(2,18)_ = 0.4772, p = 0.6291; Figure 6B). The data show an apparent increase in the expression of AHR in females compared with males in the elderly (p = 0.1316) and AD (p = 0.0768) patients. Furthermore, male AD patients displayed the highest concentration of AHR when compared with the elder male group without AD (t = 3.383, df = 14, p = 0.0045, Figure 4C). Similarly, female AD patients had a higher serum level of AHR than elder females (t = 3.015, df = 17, p = 0.0078, Figure 6C).

## 3. Discussion

In this study, we show evidence that supports, to our knowledge for the first time, the AHR association in the aging process of the human brain and, probably, in the development of AD through the response of glial cells to a pro-inflammatory environment in the central nervous system. We found an increase in the AHR expression in the brain of elder individuals and AD patients, with the most visible changes in astrocytes. Additionally, the AHR serum level was higher in AD patients compared with elderly participants. Taking into account that AHR-knockout mice are more susceptible to inflammation stimulus [20] and that the inhibition or silencing of AHR promotes an increase in the inflammatory response [11], it was expected that chronic inflammation in human aging and in AD patients would be related to a decrease of AHR expression in these conditions. Surprisingly, our results showed the opposite, since AHR in post-mortem brains and circulating AHR were higher in both aging and AD. Our results are in line with recent data informed by Rothhammer et al. [11], who reported a high AHR expression in astrocytes from post-mortem brains of patients with multiple sclerosis. In the same study, the authors suggested that the CNS-produced interferon (IFN) activates AHR in astrocytes to suppress inflammation. Taking into account that the interferon has also been implicated in either aging or AD inflammation [21], we think that the IFN–AHR axis could also be involved in the increase of AHR in aging and AD patients. Besides this, the AHR accumulation can also be triggered by the low levels of ARNT previously reported during aging. ARNT is essential for the transcriptional activity of AHR, as it depends on the binding of ARNT. Therefore, a lack of ARNT restricts the AHR localization to the cytosol, thereby inhibiting its transcriptional activity [22].

Regarding the cell-type location, the AHR protein was located in astrocytes. This finding can be explained by the fact that astrocytes are among the principal cells that participate in the modulation of inflammation during aging. We observed the AHR protein mainly in the cytosol and not in the cell nuclei, which suggests that during human aging and in AD, AHR has an impediment to act as a transcription factor. Also, in AD post-mortem brain tissue, we observed AHR with a morphology that resembles extracellular microvesicles (which can be exosomes or ectosomes, among others). Our findings using post-mortem tissue are consistent with our evidence about serum AHR, since AHR could only be detected in the detergent-treated blood serum, which suggests that AHR is circulating through the bloodstream in microvesicles derived from the brain or other tissues. The microvesicles, such as exosomes, participate in cellular communication mechanisms, with the ability to cross the blood–brain barrier allowing the circulation through the bloodstream. The molecular mechanisms involved in cell–to–cell communication through exosomes that transport proteins associated with neuropathologies can not only be used as a molecular marker of diseases, but can also provide a molecular pathway to identify and study the fundamental pathological changes of diseases like AD, from a different approach [23]. The presence of exosomes in AD is usual since previous reports demonstrated that astrocytes stimulated with β-amyloid release exosomes enriched with molecules relevant for AD, like protease-activated receptor-4 and ceramide [24]. Surprisingly, we observed that serum AHR decreases considerably in patients with AD aggravated with cerebral microangiopathy. More evidence is required to discuss this observation that undoubtedly generates significant interest.

The AHR and the astrocytes are critical elements as mediators of diverse biological processes, such as the regulation of the immune system homeostasis in response to environmental toxicants [25] or in response to infections [26], and both events are involved in different AD pathogenesis theories. Therefore, continuing with the study from the biological and physiological responses between AHR and astrocytes in aging and neurodegenerative disorders can be helpful to explain complex diseases like AD (Figure 5). In addition, further validation of the findings in larger clinical studies is needed to know the role of arylhydrocarbons and AHR in aging or neurodegenerative diseases.

In conclusion, the AHR participates in the aging process and probably in the development of AD through the response of astrocytes, which can release AHR into microvesicles. Our findings open a gap in the knowledge about the AHR regulation in the human neurodegenerative pathology, its association with glia, and the possibility that the AHR detected in the serum of patients with AD comes from the brain and is released into extracellular microvesicles (Figure 7). 

## 4. Materials and Methods 

### 4.1. Study Design and Participants

This is a case-control study, in Mexican participants, that was performed in two phases. For the first one, we studied post-mortem hippocampus tissue from young (20–30 years old, n = 7) and elder (>60 years old, n = 6) subjects of either sex, without infectious, inflammatory, or neoplastic systemic disease and macroscopic abnormalities in the brain. These samples were obtained from the Hospital General de México “Eduardo Liceaga”. We also studied post-mortem hippocampus tissues from AD patients of either sex (n = 3) that were obtained from the Brain Bank-LaNSE CINVESTAV-IPN, Mexico. 

In the second phase of this study, we recruited young (20–30 years old, n = 14) and elder (> 60 years old, n = 16) healthy participants of either sex without cognitive impairment. We also recruited AD patients of either sex (>60 years old, n = 29) with the diagnosis of probable AD dementia determined according to the National Institute of Aging-Alzheimer’s Association (NIA-AA) criteria [27] from the Instituto Nacional de Neurología y Neurocirugía of Mexico City (INNN) in 2017–2018. Global cognitive function was assessed with the MiniMental State Examination, by considering cognitive impairment in scores below 24. The dementia severity was assessed using the Clinical Dementia Rating (CDR) to sort participants and patients in the following groups: no cognitive impairment (CDR 0), mild AD (CDR 1), moderate AD (CDR 2), or severe AD (CDR 3). MRI in the INNN’s clinical records identified AD patients that presented microangiopathy. Methodological procedures agreed with the Declaration of Helsinki for experiments involving humans. The trial was approved by the Clinical Research Ethics Committee of the Hospital General de México (08 Sep 2015) with the number #DI/15/310/50 and by the Clinical Research Ethics Committee of the INNN (21 Jan 2016) with the number 116/15. All participants included in this study provided informed consent, and their security data were protected. Additional characteristics for each group are described in Table 1 and Table 2.

### 4.2. Histochemical Analysis and Immunohistochemistry 

Samples obtained from Hospital General de México were post-fixed in 4% formaldehyde, embedded in paraffin, and sectioned to obtain slices of 5 µm. Then, we analyzed the morphology trough the hematoxylin–eosin staining. We received the samples from the Brain Bank-LaNSE as slices of 30 µm. These slices were from hippocampal tissue fixed with 4% paraformaldehyde and cut in freezing conditions. Sections were washed with PBS and incubated with citrate buffer pH = 6 for antigenic retrieval, then they were washed and incubated with 0.3% H_2_O_2_ for immunohistochemistry or with 7% Sudan Black in 70% ethanol for immunofluorescence. After washing, the tissue was blocked with 5% ASB with 0.2% triton by one h and incubated overnight with primary antibodies against AHR (monoclonal MA1-514-Invitrogen, Carlsbad, CA, USA) or glial fibrillary acidic protein (GFAP, Z0334-DAKO, Real Carpinteria, CA, USA). After washing the primary antibody, sections were incubated with a secondary antibody, Goat anti-Mouse (626520-Invitrogen, Carlsbad, CA, USA) or Anti-Rabbit (C02-7074S - Cell Signalling, Danvers, MA, USA) for immunohistochemistry, and Alexa 488 and Alexa 568 for immunofluorescence, washed and mounted using Entellan for immunohistochemistry or Vectashield with DAPI for immunofluorescence. Slides were examined with a Nikon Eclipse 801 (Nikon, Corporation, Japan). 

We quantified the data through the Image J software (Rasband, 1997–2016, version 1.45) using three sections of each hippocampus sample from young (n = 7) and elder (n = 6) donors. From each section, we took five micrographs of adjacent regions in the CA1 and CA3 hippocampal regions in a total area of 17 mm^2^. To discriminate between neuron and non-neuron cells, we manually counted the stained nuclei, taking into account size and morphology using decision-making algorithms and descriptions previously reported [28]. We quantified GFAP as the total area stained, and AHR as the number of stained cells/number of total cells. The results are shown as an average of the values obtained. To test the cell type in which AHR was expressed, double immunofluorescence for AHR/GFAP was performed. Images were captured with a Leica TCS SP8 confocal microscope (Leica Microsystems, Wetzlar, Germany). We show a representative micrograph taken from the observation of one section for AD patients (n = 3).

### 4.3. Blood Collection and Processing

Blood samples were collected and processed to obtain blood serum according to the NIA-AA guidelines for the standardization of preanalytical variables in the study of blood-based biomarkers in AD [29]. Participants were requested to fast for at least eight hours before sample collection.

### 4.4. Total Protein and AHR Quantification

Serum protein levels were determined using the Lowry assay. Tween (4% *v/v*)-treated serum samples [19] were used to quantify serum AHR levels using a sandwich enzyme-linked immunoabsorbent assay (ELISA; MBS702604, My Biosource, Inc., San Diego, CA, USA) following the manufacturer´s instructions. This kit provides standards to build a calibration curve and to test a sample of known concentration. We read the plate with a multimodal microplate reader (Sinergy HT) coupled to the Gen5 v3 software (Biotek Instruments, Winooski, VA, USA), which generated the four-parameter logistic (4-PL) curve-fit needed to calculate the AHR concentration.

### 4.5. Statistical Analysis

All data are expressed as the mean ± SEM. We carried out comparisons by Unpaired Student´s t-tests or one-way ANOVA, followed by a Bonferroni test as appropriate, using the Prism 5.02 software (GraphPad, San Diego, CA, USA).

## Figures and Tables

**Figure 1 ijms-21-01983-f001:**
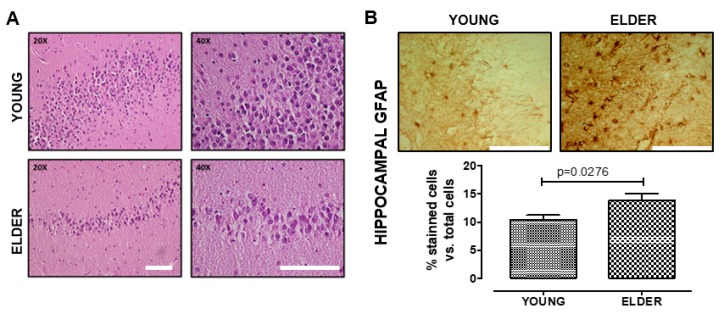
Reactive gliosis increases in the old hippocampus. We determined the hippocampal tissue integrity by hematoxylin–eosin staining and no detection of protein inclusions, cells with apoptotic characteristics, or other abnormalities (**A**). We observed a significant increase of the glial fibrillary acidic protein (GFAP) protein level in the astrocytes from the elderly vs. the young brain (**B**). Values are mean ± SEM; Statistics were performed by unpaired t-test. Young (*n* = 7): 20–30 years old. Elder (*n* = 6): >60 years old. Scale bar = 100 μm.

**Figure 2 ijms-21-01983-f002:**
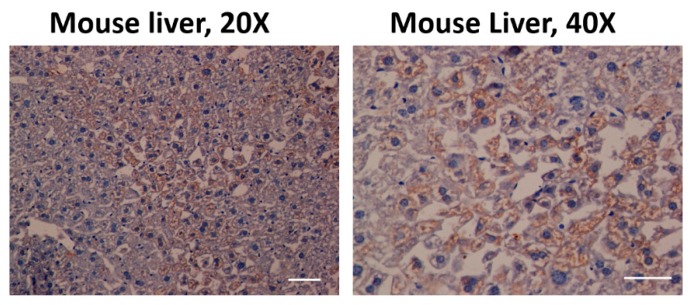
Aryl hydrocarbon receptor (AHR) expression in C57BL/6 mouse liver. The antibody used against AHR was effective (coffee stain). Likewise, the samples were also stained with hematoxylin (blue) to contrast the cell nuclei Scale bar = 100 μm.

**Figure 3 ijms-21-01983-f003:**
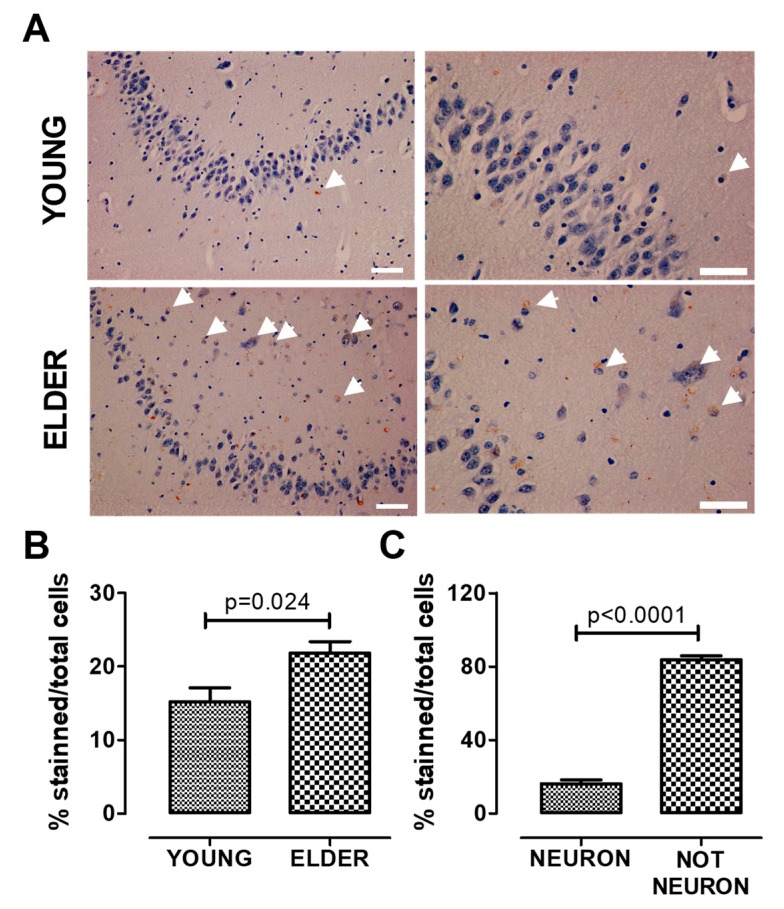
AHR expression increases in the old hippocampus. We determined AHR expression by immunostaining and found that AHR expression increases in the elderly. The higher expression was found in non-neuronal cells. Representative images (**A**), total quantification (**B**), and AHR staining quantification in neurons and non-neuronal cells from the elderly (**C**). Arrows show the AHR staining. Values are mean ± SEM; Statistics were performed by unpaired t-test. Young (n = 7): 20–30 years old. Elder (*n* = 6): >60 years old. Scale bar = 50 μm.

**Figure 4 ijms-21-01983-f004:**
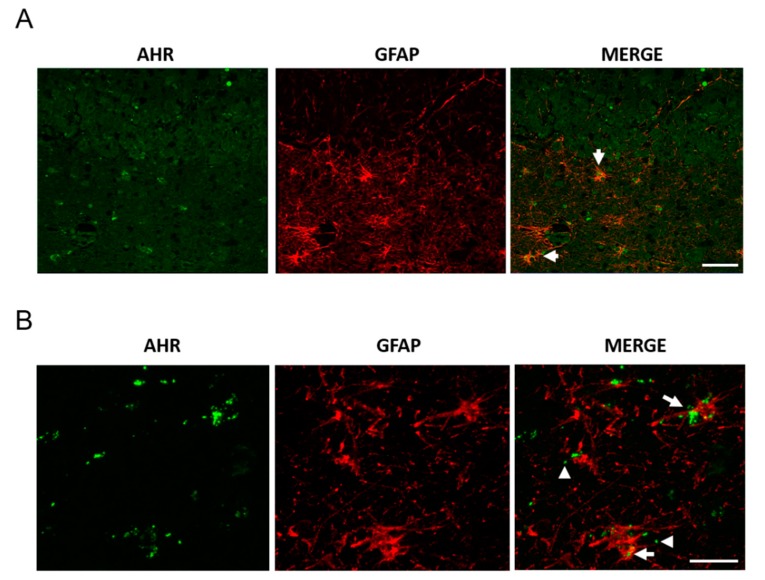
AHR and GFAP colocalization in post-mortem tissue. We determined AHR and GFAP expression by immunostaining and found AHR in astrocytes, apparently as microvesicles. Representative images from a 94 year-old female donor (**A**, scale bar = 50 μm) and a 100 year-old Alzheimer’s disease (AD) female donor (**B**, scale bar = 20 μm). The arrows show the AHR staining in astrocytes. The arrowheads show AHR with the appearance of microvesicles. We analyzed samples from three donors.

**Figure 5 ijms-21-01983-f005:**
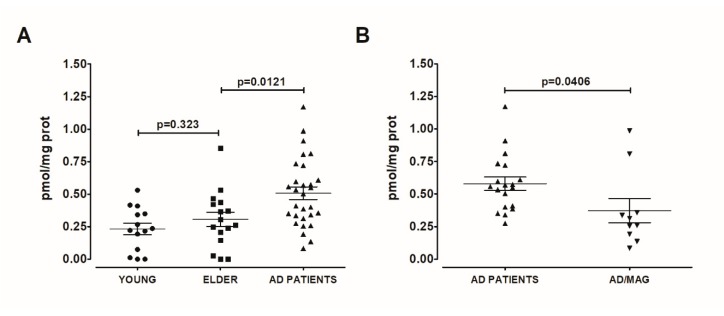
AHR increase in AD blood serum. We determined AHR in serum from the young, healthy elder, and AD patients. We found higher AHR concentration in AD patients (*n* = 29) than in young (*n* = 14) and elderly (*n* = 16) participants (**A**); microangiopathy (MAG, *n* = 10) is a comorbidity that decreases the AHR level in AD patients (**B**). Values are mean ± SEM. Statistics were performed by unpaired t-test. We did not detect significant differences in variances using an F-test.

**Figure 6 ijms-21-01983-f006:**
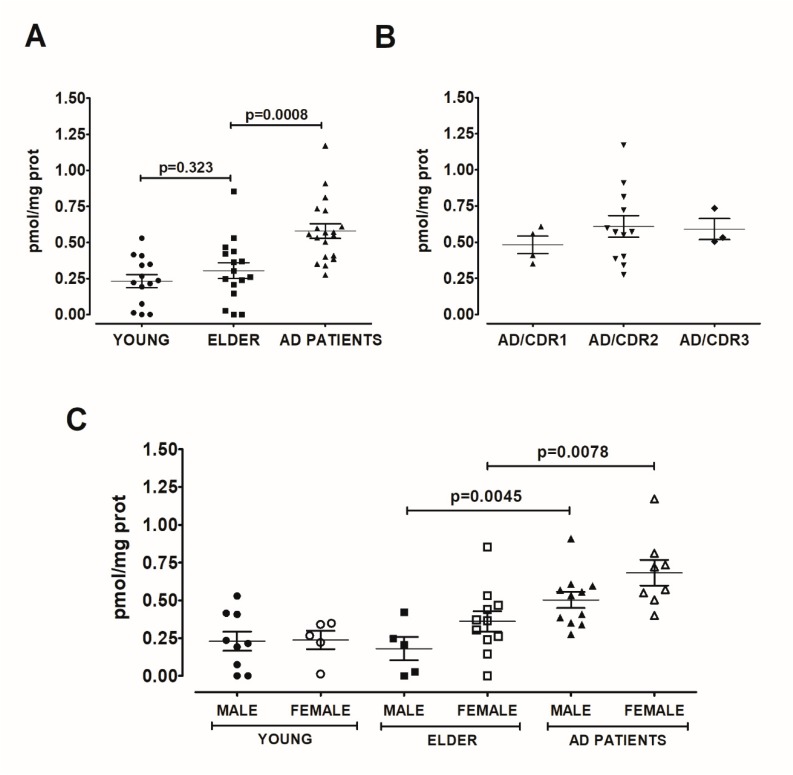
The AHR increase did not depend on AD severity or sex. As microangiopathy decrease the AHR serum concentration, we analysed only the AD patients without this comorbidity (**A**). We found that AHR concentration did not depend on AD severity (**B**) or sex (**C**). Values are mean ± SEM. Statistics were performed by unpaired t-test in (**A**) and (**C**) and by one-way ANOVA followed by Bonferroni test in (**B**). We did not detect significant differences in variances using an F-test for the t-test and a Bartlett test for one-way ANOVA. Young (n = 14): 20–30 years old. Elder (n = 16): >60 years old. AD (*n* = 19): Patients with AD, without microangiopathy. The dementia severity was assessed using the Clinical Dementia Rating (CDR) to sort Alzheimer’s disease patients in the following groups: mild AD (CDR1), moderate AD (CDR2), or severe AD (CDR3).

**Figure 7 ijms-21-01983-f007:**
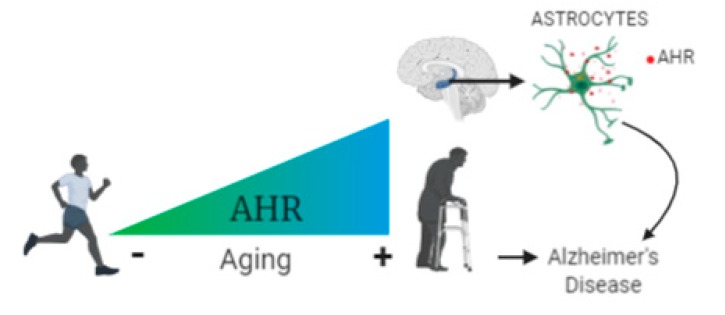
AHR is associated with the aging process and probably with the development of AD (Alzhimer’s Disease) through the response of hippocampal astrocytes, which can release AHR into microvesicles.

**Table 1 ijms-21-01983-t001:** Characteristics of post-mortem brain tissue.

Sex	Age	Group
Female	18	Young
Female	26	Young
Female	27	Young
Female	30	Young
Male	23	Young
Male	28	Young
Male	30	Young
Female	65	Elder
Female	89	Elder
Female	94	Elder
Male	70	Elder
Male	72	Elder
Male	75	Elder
Female	67	AD
Female	100	AD
Male	60	AD

Individuals between 20–30 and > 60 years had a record and cause of death defined. Exclusion criteria: systemic infection, inflammation or neoplasia, macroscopic abnormalities in the brain or neurological diseases (except donors with AD).

**Table 2 ijms-21-01983-t002:** Characteristics of the serum samples studied.

Group (n)	Female/Male (%)	Median Age ±SD (Years)
Young (n = 14)	36/64	24.5 ± 2.1
Elder (n = 16)	3/69	77.7 ± 3.7
Total AD patients (n = 29)	55/45	73.6 ± 7.2
AD patients without microangiopathy (n = 19)	42/58	73 ± 7
AD patients with microangiopathy (n = 10)	80/20	74.7 ± 7.2

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
