# Peer review of "Aryl Hydrocarbon Receptor in Post-Mortem Hippocampus and in Serum from Young, Elder, and Alzheimer’s Patients"

_ijms, 2020, doi:10.3390/ijms21061983_

Round 1

Reviewer 1 Report

  1. The overexpression of AHR is associated with ageing and more prominently with Alzheimer's disease.  The data presented do not suggest or prove AHR involvement in ageing or Alzheimer's diseases as authors have written.  Consider revising that language: there is only an association, but sample size is small.  As presented, the study didn't explore how AHR is involved in ageing or causing Alzheimer's disease. AHR mediated inflammation, cell senescence and necrosis may be mechanisms, but we do not know that for sure.   
  2. Sample size is small but the results are encouraging.  Further validation of the findings in larger observational, cohort or case/control studies is needed to assign a role to arylhydrocarbons (such as PAHs nitosamines, tobacco carcinogens, etc) and AHR in ageing or neurodegenerative diseases.
  3. It is difficult to see that the AHR is localized in vesicles.  Electron Microscopy studies may help to make this clear.    

Author Response

Ref: ijms-736822

Title: Aryl Hydrocarbon Receptor in post-mortem hippocampus and in serum from young, elders and Alzheimer's patients

Journal: International Journal of Molecular Sciences

Response to Reviewer 1 Comments

We appreciate your comments, which help us to improve our manuscript

Point 1. The overexpression of AHR is associated with ageing and more prominently with Alzheimer's disease.  The data presented do not suggest or prove AHR involvement in ageing or Alzheimer's diseases as authors have written.  Consider revising that language: there is only an association, but sample size is small.  As presented, the study didn't explore how AHR is involved in ageing or causing Alzheimer's disease. AHR mediated inflammation, cell senescence and necrosis may be mechanisms, but we do not know that for sure.   

R1. We agree with the reviewer; our data do not support an AHR involvement in ageing or Alzheimer's diseases, so we changed the term “involvement” by “association” inline 209.

Point 2. The sample size is small, but the results are encouraging. Further validation of the findings in larger observational, cohort or case/control studies is needed to assign a role to aryl hydrocarbons (such as PAHs nitosamines, tobacco carcinogens, etc) and AHR in ageing or neurodegenerative diseases.

R2. As this comment is particularly relevant for our research field, we added the reviewer’s idea in our discussion, lines 258-259.

Point 3. It is difficult to see that the AHR is localized in vesicles. Electron Microscopy studies may help to make this clear. 

R2. This concern is particularly relevant; in fact, we are currently working to study extracellular vesicles in serum. As the first step, we will perform electron microscopy studies, but now we do not have data about it. 

Reviewer 2 Report

In the present study, the authors evaluated the changes in the expression of the AHR and GFAP in human hippocampal post-mortem tissues from young and elder subjects. They also assessed AHR levels in serum from young and elder participants as well as Alzheimer's patients. The findings of this study were that the levels of AHR and Glial Fibrillar Acid Protein were higher in elder than in young post-mortem brain samples; AHR localized mainly in the cytosol of astrocytes and displayed a pattern that resembles extracellular vesicles; this last feature was more conspicuous in AD subjects; higher serum levels of AHR in AD patients than the other participants. This study is interesting but opposite to the previous study of Rothhammer et al the authors suggested. However, the increased level of AHR may be a secondary response for anti-inflammation in AD’s patients because the serum AHR was not dependent on disease severity and serum AHR decreased in AD’s patients aggravated with cerebral microangiopathy. Otherwise, the level of serum AHR should be increased in AD’s patients aggravated with cerebral microangiopathy. In figure 4A, serum AHR in AD’s patients with cerebral microangiopathy was higher than AD’s patients without microangiopathy. I have some concerns and recommendations about this study as follows.

  1. Figure 2C showed AHR staining quantification in neurons and non-neuronal cells. AHR expression increases in non-neuronal cells. It means young people, elder people or all of the samples? How were the neuron and non-neuron differentiated? Was there any cell marker to be detected? How can the authors say “Regarding cell type location, the AHR protein was predominantly located in astrocytes”?
  2. In figure 3, only two female donors were demonstrated. How about the male? Were they the same results? We can find some difference of AHR level in serum between male and female either elder or AD patients (Fig 4C).
  3. In Fig 4A, AD PATIENTS or AD/without MAG (n=19)? And the statistics should be adjusted by sex due to uneven distribution among these groups. Serum AHR level in elder should be compared with total AD’s patients.
  4. In Fig 4B, one way ANOVA could be not adequate due to the sample size. How about the normal distribution and homogeneity of variance of these data?
  5. In Fig 4C, were significantly different between male and female either elder or AD patients in statistics? If it is significant, the sex should be adjusted.
  6. Aryl hydrocarbon receptor (AhR) is widely expressed by different cell types throughout the body. How did link the serum AHR with astrocytes? How were the microvesicles released into blood?

Rothhammer, V.; Mascanfroni, I.D.; Bunse, L.; Takenaka, M.C.; Kenison, J.E.; Mayo, L.; Chao, C.C.; Patel, B.; Yan, R.; Blain, M., et al. Type I interferons and microbial metabolites of tryptophan modulate astrocyte activity and central nervous system inflammation via the aryl hydrocarbon receptor. Nat Med 2016, 22, 586-597, doi:10.1038/nm.4106. (作者有引用)

Author Response

Ref: ijms-736822

Title: Aryl Hydrocarbon Receptor in post-mortem hippocampus and in serum from young, elders and Alzheimer's patients

Journal: International Journal of Molecular Sciences

Response to Reviewer 2 Comments

We appreciate your comments, which help us to improve our manuscript

Point 1. Figure 2C showed AHR staining quantification in neurons and non-neuronal cells. AHR expression increases in non-neuronal cells. Does it mean young people, elder people or all of the samples? How were the neuron and non-neuron differentiated? Was there any cell marker to be detected?

R1. We quantify the AHR staining in neuron and not neuron cells from elder brains. To clarify this, we modified the figure legend of figure 3, lines 133-136. We use nucleus morphology and size as the basis to discriminate between the neuron and not neuron cells, according to García-Cabezas et al. (2016). We added this information to the Material and methods section, line 337-339, and the mentioned reference was also added.

Point 2. How can the authors say, “Regarding cell type location, the AHR protein was predominantly located in astrocytes”?

R2. We declared that AHR protein was predominantly located in astrocytes because we found AHR and GFAP colocalization (Figure 4). As we are aware that AHR could be expressed in other cells, we modify the sentence mentioned by the reviewer to state it, line 233.  

Point 3. In figure 3, only two female donors were demonstrated. How about the male? Were they the same results? We can find some difference of AHR level in serum between male and female either elder or AD patients (Fig 4C).

R3. We found a significant increase of AHR expression in elder's brains, but this was not different by sex, so we do not expect differences by sex in AD donors. We decided to show data from females because they were from the eldest donors. The difference between male and female that is shown in figure 6C is only apparent, as there is no statistical difference. As we have not included the p-value to the elderly, we added it in line161-162.

Point 4. In Fig 4A, AD PATIENTS or AD/without MAG (n=19)? And the statistics should be adjusted by sex due to uneven distribution among these groups.

R4. Nineteen is the number of AD without MAG patients, but we did not state it because we consider more relevant to mention the presence of comorbidity that to clarify its absence. Although the data for men and women are shown in the same graph, the statistical analysis was performed by unpaired t-test comparing each group of males with their paired female, and we did not detect significant differences in variances using F-test.

Point 5. Serum AHR level in elder should be compared with total AD’s patients.

R5. At first, we did compare elders with a total of AD patients, but we decided to separate the last considering the presence of microangiopathy. However, to increase clarity considering the reviewer's comment, we restructured figure 4 into figure 4 and 5, line 166 to 207.

Point 6. In Fig 4B, one way, ANOVA could be not adequate due to the sample size. How about the normal distribution and homogeneity of variance of these data?

R6. We agree with the reviewer, the sample size is small and uneven. However, we tested these groups using Bartlett’s test and did not find significant differences in variances. However, we performed Kruskal-Wallis test and confirmed the absence of significant differences (p=0.7819, Kruskal-Wallis statistic=0.4921). 

Point 7. In Fig 4C, were significantly different between male and female either elder or AD patients in statistics? If it is significant, the sex should be adjusted.

R7. We did not find differences, please see R4.

Point 8. Aryl hydrocarbon receptor (AhR) is widely expressed by different cell types throughout the body. How did link the serum AHR with astrocytes? How were the microvesicles released into blood?

R8. We agree with the reviewer, AHR can be from any tissue, including the brain. Thus, we did not assure that circulating AHR comes from the brain. However, we are sure that circulating AHR is in microvesicles because it was almost undetectable in blood serum. The only way to measure circulating AHR was the previous treatment with detergent, a reported way to break circulating exosomes. We think that AHR could come from the brain because cerebral comorbidity modifies the serum AHR level. Also, the examination of astrocytes under a confocal microscopy show AHR in apparent extracellular vesicles. All this information is currently in the manuscript.

Reviewer 3 Report

Regarding the paper "Aryl Hydrocarbon Receptor in post-mortem hippocampus and in serum from young, elders and Alzheimer's patients" the contents are innovative and present a new perspective on the role of the AHR in the development of Alzheimer's disease and possibly other pathologies related with stress and chronic accumulation of misfolded proteins.

Some small concerns are here pointed out:

  • Line 61 & 62: add acronyms in front of both factors for cell development.
  • Line 108: would enrich the paper if this absent data could be attached.
  • In Figure 4B would be very interesting to observe if any differences do occur in each set of AD severity according to gender (divide each group of severity by gender).
  • Line 202: you state "observational and case-control study" but a case-control study is included in the observational studies.

The chapter of Discussion is very well achieved. Nevertheless, according to the recent literature, I would like to see this topic further extended regarding the potential of exosomal signaling in the context of AD development and chronic stress settlement.

Overall, I classify this paper as an attractive article in the field. 

Author Response

Ref: ijms-736822

Title: Aryl Hydrocarbon Receptor in post-mortem hippocampus and in serum from young, elders and Alzheimer's patients

Journal: International Journal of Molecular Sciences

Response to Reviewer 3 Comments

Regarding the paper "Aryl Hydrocarbon Receptor in post-mortem hippocampus and in serum from young, elders and Alzheimer's patients" the contents are innovative and present a new perspective on the role of the AHR in the development of Alzheimer's disease and possibly other pathologies related with stress and chronic accumulation of misfolded proteins.

Point 1. Line 61 & 62: add acronyms in front of both factors for cell development.

R1. We added the suggested acronyms, line 61-62.

Point 2. Line 108: would enrich the paper if this absent data could be attached.

R2. We added the data suggested in a new figure. In consequence, the number of the following graphs was changed. These changes are in line 116-120.

Point 3. In Figure 4B would be very interesting to observe if any differences do occur in each set of AD severity according to gender (divide each group of severity by gender).

R3. We agree with the reviewer about this interesting observation. However, the suggested comparison is not possible because of the small sample size.

Point 4. Line 202: you state "observational and case-control study" but a case-control study is included in the observational studies.

R4. Thank you for your observation, we changed the description to “This is a case-control study”, line 298.

Point 5. The chapter of Discussion is very well achieved. Nevertheless, according to the recent literature, I would like to see this topic further extended regarding the potential of exosomal signaling in the context of AD development and chronic stress settlement.

R5. We add a paragraph in the discussion addressing your suggestion Lines 237-249

Point 6. Overall, I classify this paper as an attractive article in the field. 

R6. Thank you for your valuable comments